# Femtosecond X-ray induced changes of the electronic and magnetic response of solids from electron redistribution

Daniel J. Higley [1,2]*, Alex H. Reid [1], Zhao Chen[1,3], Loïc Le Guyader[1,4], Olav Hellwig[5,6,7], Alberto A. Lutman [1], Tianmin Liu[1,3], Padraic Shafer [8], Tyler Chase[1,2], Georgi L. Dakovski[1], Ankush Mitra[1,9], Edwin Yuan[1,2], Justine Schlappa[4], Hermann A. Dürr[1,10], William F. Schlotter[1] & Joachim Stöhr[1]*

Resonant X-ray absorption, where an X-ray photon excites a core electron into an unoccupied valence state, is an essential process in many standard X-ray spectroscopies. With increasing X-ray intensity, the X-ray absorption strength is expected to become nonlinear. Here, we report the onset of such a nonlinearity in the resonant X-ray absorption of magnetic Co/Pd multilayers near the Co $L_3$ edge. The nonlinearity is directly observed through the change of the absorption spectrum, which is modified in less than 40 fs within 2 eV of its threshold. This is interpreted as a redistribution of valence electrons near the Fermi level. For our magnetic sample this also involves mixing of majority and minority spins, due to sample demagnetization. Our findings reveal that nonlinear X-ray responses of materials may already occur at relatively low intensities, where the macroscopic sample is not destroyed, providing insight into ultrafast charge and spin dynamics.

[1] SLAC National Accelerator Laboratory, 2575 Sand Hill Road, Menlo Park, CA 94025, USA. [2] Department of Applied Physics, Stanford University, Stanford, CA 94305, USA. [3] Department of Physics, Stanford University, Stanford, CA 94305, USA. [4] European X-Ray Free-Electron Laser Facility GmbH, Holzkoppel 4, 22869 Schenefeld, Germany. [5] San Jose Research Center, HGST a Western Digital Company, 3403 Yerba Buena Rd., San Jose, CA 95135, USA. [6] Institute of Physics, Chemnitz University of Technology, 09107 Chemnitz, Germany. [7] Institute of Ion Beam Physics and Materials Research, Helmholtz-Zentrum Dresden-Rossendorf, 01328 Dresden, Germany. [8] Lawrence Berkeley National Laboratory, Berkeley, CA 94720, USA. [9] Department of Physics, University of Warwick, Coventry CV4 7AL, UK. [10] Department of Physics and Astronomy, Uppsala University, Box 516, 75120 Uppsala, Sweden. *email: dhigley@slac.stanford.edu; stohr@slac.stanford.edu

With the advent of X-ray free electron lasers (XFELs) it is now possible to produce high fluence X-ray pulses of tens of femtoseconds duration that transform a solid sample into exotic states of matter with dramatically changed electronic structure and response to X-rays[1–7]. Such states, resembling a solid-density plasma, are typically observed when the electronic temperature of the sample becomes of order 10 eV and exceeds atomic bonding energies. At lower X-ray fluences more subtle changes of the electronic structure of solids will be induced, and it is of key importance to understand the associated dynamic mechanisms and their thresholds.

The transition from the conventional linear response to various kinds of non-linear effects is most directly revealed by X-ray absorption spectroscopy since the absorption process is responsible for the transfer of energy to the sample. In particular, resonant X-ray absorption spectroscopy, which measures photon energy and intensity dependent transitions of core electrons to quasi-localized empty valence states, is most sensitive to subtle changes in the electronic valence states. If carried out with circularly polarized X-rays it can furthermore separate charge and spin dependent effects. The resonant absorption fine structure is most pronounced in the soft X-ray region ($\approx$200–2000 eV), where its dependence on linear polarization has been widely used to obtain element specific information of chemical bonding in molecules and organic matter[8] and its dependence on circular polarization has provided information on the magnetic properties of transition metals and their complexes[9,10].

In resonant soft X-ray absorption spectroscopy inner shell electrons are excited into empty valence states, either in the form of molecular orbitals or empty quasi-localized band states. For 3d transition metals the all important 3d valence states are mapped out through dipole transitions from the 2p core states, giving rise to pronounced L-edge resonances. The X-ray absorption excited state decays on the 1–10 femtosecond time scale determined by the lifetime of the core hole, which is dominated by the faster Auger rather than radiative fluorescent decay channel[11].

In conventional X-ray absorption experiments, even with the most advanced synchrotron radiation sources, typically only a single atom in the sample is excited during the core hole lifetime. Hence the absorption processes are sequential, one photon at a time, and are independent of other absorption or scattering processes, described by linear response perturbation theory. In contrast, with XFELs the photon degeneracy parameter is increased from <1 for synchrotron sources to a maximum of about $10^9$ (ref. [12]) and a large number of absorption processes may occur simultaneously or within a typical pulse length in the 10–50 fs range. Interactions of X-rays with a sample may then no longer be independent and different non-linear processes may occur. The two most basic processes consist of two or more photons working together to couple the absorption and decay processes within the core hole lifetime, referred to as stimulated resonant scattering[13–16], or the modification of a given atomic absorption event by the perturbation caused by the other electronic excitations within the pulse duration. In both cases, the absorption spectrum, recorded as a statistical average over many pulses, will then be changed in shape or intensity.

Here we report the direct observation of the onset of the non-linear response of a magnetic solid through electronic charge and spin rearrangement within the valence band over a timescale of <40 fs. These effects are observed by use of X-ray magnetic circular dichroism (XMCD) absorption spectroscopy[9] measurements across the Co $L_3$ resonance (778 eV) of Co/Pd magnetic multilayers. The measurements are made with monochromatic X-rays of $\simeq$300 meV energy width and $\simeq$40 fs FWHM pulse lengths, produced at the Linac Coherent Light Source (LCLS). We observe that with increasing incident intensity, the absorption of

X-rays is increasingly modified by the femtosecond dynamic electronic response during the pulse itself, resulting in a characteristic change of the resonant absorption line shape for the time-integrated pulses. These changes occur at remarkably low deposited energy densities, reaching resonant absorption changes larger than ten percent for deposited energy densities of 280 meV per atom, corresponding to the absorption of one photon for every 3000 atoms in the sample. These deposited energy densities are orders of magnitude lower than those required for similar changes in non-resonant absorption[1,5], but similar as previously employed in diffraction imaging of similar samples, where changes in diffraction were attributed to stimulated forward scattering[15]. The shape of the absorption changes, however, are different than those predicted within the model used to describe those results[13]. We find instead, for the parameter range investigated here, that our results are described remarkably well by a simple model where the absorption changes are dominated by valence electronic and magnetic changes of the sample itself. In this model energy deposited by an X-ray photon is transferred to valence electrons within 2 eV of the Fermi level within 20 fs, and the sample demagnetizes by more than twenty percent within the X-ray pulse duration.

## Results

**Experimental setup.** To measure the fluence dependence of X-ray absorption across the Co $L_3$ edge, we used the technique developed in[17] with a few modifications to accommodate our use of high and variable incident X-ray fluence (see Methods). The key components of the experimental setup are shown schematically in Fig. 1. Circularly polarized X-ray pulses produced by the Delta undulator of the Linac Coherent Light Source (LCLS) were monochromatized to a bandwidth of 260 meV. These X-ray pulses passed through a fluorescence-based relative X-ray pulse energy detector ($I_0$)[17] before being focused onto Co/Pd magnetic multilayer samples. The samples had a metal layer sequence of Ta (1.5)Pd(3)[Co(0.6)Pd(0.6)] × 38Pd(2), where the thicknesses in parentheses are in nm, and were magnetized out-of-plane. The total energy of X-ray pulses transmitted through the samples was detected with a CCD ($I_1$). The absolute X-ray pulse energy at the sample was determined using the CCD and a gas-based detector upstream of the samples (see Methods). The photon energy of the spectra recorded with the XFEL were calibrated to align with

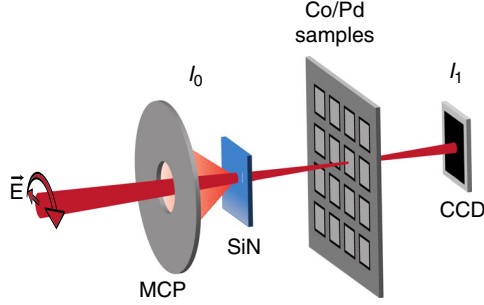

**Fig. 1** Experimental setup for X-ray absorption spectroscopy. The setup was implemented at the Linac Coherent Light Source (LCLS) X-ray Free Electron Laser (XFEL). It was used to measure X-ray absorption as a function of photon energy and relative orientation of circular X-ray polarization and sample magnetization. The LCLS Delta undulator produced circularly polarized X-rays which were then monochromatized. This resulted in X-ray pulses with 260 meV bandwidth and 39 fs FWHM duration. The total energy of these X-rays was detected with the fluorescence-based $I_0$ detector before they were focused on a Co/Pd sample. The X-ray pulse transmitted through the sample was attenuated before being detected with a CCD ($I_1$).

those measured at the synchrotron light source for the lowest measured X-ray fluences. A linear background was subtracted from the XFEL spectra such that the high fluence spectra aligned with the low fluence spectra well away from the absorption resonance (see Methods and Supplementary Fig. 1).

To calculate the X-ray excitation fluence, we have to take into account that both the excitation profile and probing profile are created by the same X-ray spot on the sample. The X-ray spot is approximately Gaussian and the X-ray fluence varies over this Gaussian spot. We quote the X-ray fluence as the fluence averaged over the X-ray pulse profile and weighted by the X-ray fluence at each point on the sample. This averaged X-ray fluence is one half of the peak X-ray fluence.

**Dependence of X-ray absorption on incident X-ray fluence.**
Figure 2 shows the dependence of the absorption spectra around

the Co $L_3$ resonance on incident X-ray fluence. At this resonance, X-ray absorption excites Co $2p_{3/2}$ core electrons into unoccupied valence states which are primarily of 3d character. Spectra were recorded in the low fluence limit at the Advanced Light Source (ALS) synchrotron light source[18] (dashed spectra labeled ALS), and compared to those recorded with variable incident X-ray fluence at LCLS. In Fig. 2, XAS refers to the X-ray absorption spectrum found by averaging the X-ray absorption spectra measured with opposite orientations of X-ray polarization and sample magnetization. The correct photon energy for exciting Co $2p_{3/2}$ core electrons into unoccupied states at the Fermi level is estimated as the zero crossing of the change in XAS, between where the change in XAS is positive below the absorption resonance, and negative at the peak of the absorption resonance. The reasoning for this is described in the discussion section below. We refer to this excitation energy as $E_{core \rightarrow Fermi}$, and its position is indicated in Fig. 2 with dashed vertical lines at 777.5 eV. The incident X-ray photon energy is shown on the bottom x-axes of Fig. 2, while the photon energy above $E_{core \rightarrow Fermi}$ is shown on the top x-axes. Above $E_{core \rightarrow Fermi}$, the incident X-rays have sufficient photon energy to excite Co $2p_{3/2}$ core electrons into unoccupied valence states.

Figure 2a shows, for the low fluence limit (ALS) and 43 mJ/cm² incident fluence cases, the dependence of X-ray absorption intensity on photon energy and the relative orientation of X-ray polarization and sample magnetization. One case of relative orientation of sample magnetization and X-ray polarization ($\sigma_-$) gives a large absorption resonance due to efficient excitation into the unoccupied states with the dominant spin orientation, while the other ($\sigma_+$) gives a much weaker absorption resonance. The Pd component of the samples contributes most of the non-resonant X-ray absorption intensity (an X-ray absorption intensity of 0.315 calculated using[19]).

Usually, spectra such as those shown in Fig. 2a are processed into and shown as so-called XAS (X-ray Absorption Spectrum) and XMCD (X-ray Magnetic Circular Dichroism) spectra. The XAS spectrum is the average of the $\sigma_+$ and $\sigma_-$ spectra, while the XMCD spectrum is their difference. In the low fluence limit, the XAS spectrum reflects the sample's electronic structure, while the XMCD spectrum reflects its magnetic structure[9]. The division of the lowest and highest fluence spectra into these components is shown in Fig. 2b. These spectra are calculated using the same data as that of Fig. 2a, and are simply a different illustration of the data.

Figure 2c, d shows the difference between X-ray absorption measured at LCLS and that at ALS for varying incident X-ray fluences at LCLS. Figure 2c shows this for the different orientations of X-ray polarization and sample magnetization ($\sigma_+$ and $\sigma_-$), while Fig. 2d shows this for the XAS and XMCD. For incident X-ray fluences up to 2.7 mJ/cm², there are no spectral changes outside of the noise of the measurement. For an incident X-ray fluence of 43 mJ/cm², X-ray absorption changes of more than ten percent of the resonant X-ray absorption magnitude occur. The XAS (Fig. 2b and d) increases with increasing incident X-ray fluence below $E_{core \rightarrow Fermi}$ (<777.5 eV), while it decreases with increasing incident X-ray fluence above $E_{core \rightarrow Fermi}$ (>777.5 eV). The XAS changes are localized to within 2 eV of $E_{core \rightarrow Fermi}$ and do not extend over the entire absorption resonance. The XMCD decreases with increasing incident X-ray fluence at a faster percentage rate than the changes in XAS (note scaling factor of 2 in Fig. 2d). The decrease in XMCD extends to higher photon energies than the decrease in XAS.

**Dependence of X-ray absorption on deposited X-ray energy.**
The spectra of Fig. 2 are shown for different incident X-ray

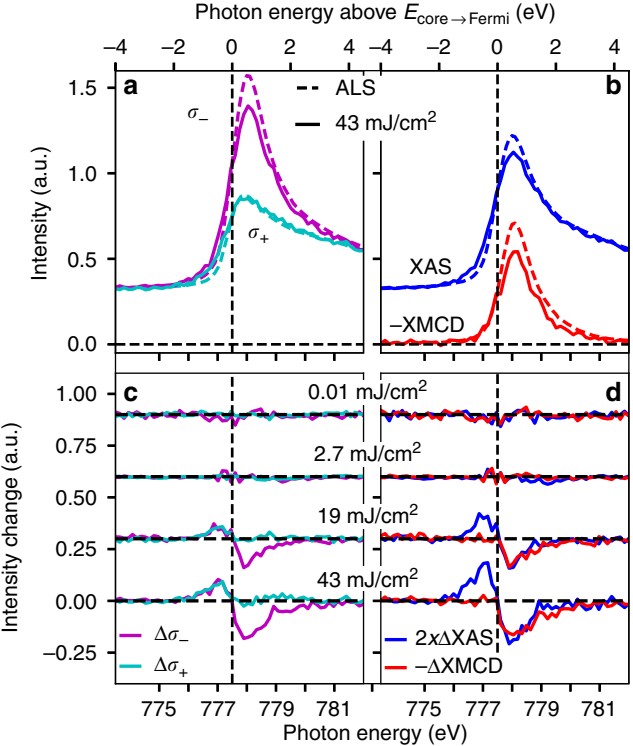

**Fig. 2** Dependence of X-ray absorption on incident X-ray fluence. Absorption spectra were recorded in the low fluence limit at the Advanced Light Source (ALS) synchrotron light source, and with variable incident X-ray fluence at the Linac Coherent Light Source (LCLS). The spectra are shown in their original recorded forms with parallel ($\sigma_-$) and anti-parallel ($\sigma_+$) orientations of X-ray polarization with respect to sample magnetization **a, c** in addition to forms derived from these **b, d**. In parts **b** and **d**, XAS is the X-ray Absorption Spectrum found by averaging the $\sigma_-$ and $\sigma_+$ spectra, while XMCD is the X-ray Magnetic Circular Dichroism spectrum found by subtracting the $\sigma_-$ spectrum from the $\sigma_+$ spectrum. The correct photon energy for exciting Co $2p_{3/2}$ core electrons into unoccupied states at the Fermi level was estimated as the zero crossing of the fluence-dependent XAS changes. This photon energy is indicated with dashed vertical lines and labeled $E_{core \rightarrow Fermi}$. **a** X-ray absorption spectra recorded with parallel ($\sigma_-$) and anti-parallel ($\sigma_+$) orientations of X-ray polarization and sample magnetization. **b** XAS and XMCD spectra calculated from the data shown in **a**. **c** Difference of X-ray absorption spectra recorded at varying incident X-ray fluence at LCLS relative to those recorded in the low fluence limit. **d** Difference of XAS and XMCD recorded with varying incident X-ray fluence at LCLS relative to the low fluence limit. Source data provided as a Source Data file.

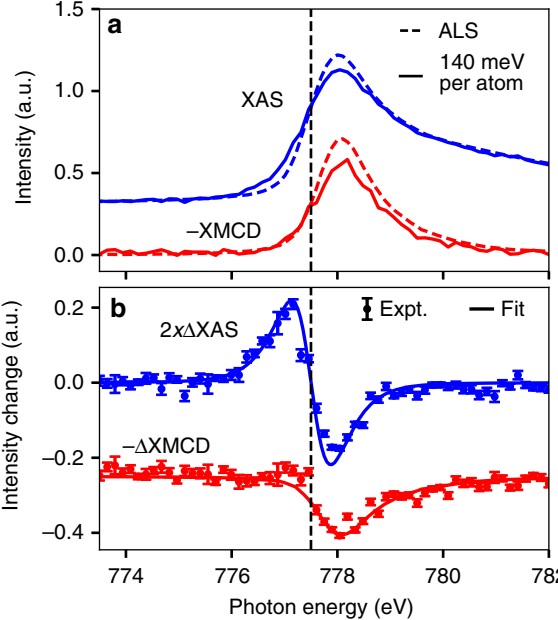

**Fig. 3** Dependence of X-ray absorption on deposited X-ray energy. This figure shows spectra recorded with 140 meV/atom pulse averaged X-ray energy absorbed in the Co/Pd multilayer (280 meV/atom total absorbed energy, 16 mJ/cm² total absorbed fluence) and those recorded on the same sample in the low fluence limit at the Advanced Light Source (ALS) synchrotron light source. **a** XAS and XMCD. **b** Changes in XAS and XMCD relative to the low fluence limit. The XAS change is fit to that expected for an electronic temperature change described by a Fermi-Dirac distribution with a constant density of states near the Fermi level, while the XMCD change is fit to a uniform reduction in XMCD. The vertical error bars show the standard error of the displayed quantities. Each combination of photon energy and sample magnetization direction had a median of 11 measurements with different X-ray pulses. Source data provided as a Source Data file.

fluences, but the degree of sample excitation scales with the absorbed X-ray energy density rather than the incident X-ray fluence. Because of this, the manner of plotting in Fig. 2 may not be ideal. In Fig. 2 each spectrum is calculated with an approximately constant X-ray fluence incident on the sample. The percentage of the incident fluence that is absorbed in the sample, however, varies with photon energy and relative orientation of X-ray polarization and sample magnetization. Thus, for each spectrum shown in Fig. 2 the degree of sample excitation varies with these parameters. A better way to plot this could be to show spectra that have been calculated to have an approximately constant absorbed fluence rather than incident fluence by varying the incident fluence as a function of photon energy and sample magnetization. Such spectra are shown in Fig. 3.

In this figure the spectral changes represent the state of the sample averaged over the X-ray pulse duration. The deposited X-ray energy averaged over the X-ray pulse duration is one half of the total deposited X-ray energy. We therefore show the spectral changes with respect to the pulse averaged absorbed X-ray energy density (absorbed X-ray fluences were converted to absorbed X-ray energies per atom using the atomic densities given in ref. [20]). The XAS changes observed below and above $E_{core \rightarrow Fermi}$ are approximately equal and opposite in shape and magnitude. The XMCD in Fig. 3b is reduced in strength relative to its low X-ray fluence limit. Comparing this to a scaled version of the XMCD measured in the low fluence limit (fit) reveals no XMCD line shape changes outside of the experimental error.

The results of Fig. 3 show that for experiments that use femtosecond X-ray pulses to measure the linear, undamaged response of solids to X-ray pulses the maximum permissible X-ray fluence can be remarkably low. The resonant X-ray absorption of Co/Pd changes by more than ten percent with a total deposited X-ray energy density of 280 meV/atom. Thus, even for experiments which seek to measure linear resonant X-ray absorption with an accuracy of only 10%, the deposited X-ray energy density should be much <280 meV/atom. This dose is much lower than the estimated sample melting dose of 615 meV/atom (see methods), which sets another limit on the maximum permissible X-ray fluence in cases where samples are not replaced between measurements.

## Discussion

As we will show, the observed X-ray absorption changes are explained by electronic and magnetic changes of the sample induced by the dynamics following absorption of X-rays. The absorption of soft X-ray photons in a solid leads to the creation of high energy photoelectrons from outer shells with binding energies less than the photon energy and core holes that are filled on the femtosecond time scale by electrons from outer shells, mostly through Auger processes. For the resonantly excited core shell, the created electron is transiently trapped in empty localized valence states and may participate in filling the core hole or delocalize into band-like states[21]. Decay of the core hole proceeds on the timescale of the core hole lifetime. In the soft X-ray region the lifetime of 1–10 fs is determined by the faster non-radiative Auger decay rather than the slower radiative fluorescence decay[11]. Auger decay initiates an electron cascade through inelastic electron-electron collisions. While scattered electrons with energies above about 5 eV may overcome the surface potential barrier (work function) and escape into vacuum, most electrons (>90%) cascade down to energies ≲5 eV in ≈10 fs and spread over nanometers[22,23].

Despite its importance, the dynamics of scattering events for electrons with kinetic energies in the 5–100 eV range is difficult to disentangle experimentally and is a matter of ongoing research[24,25]. The low energy electrons cause transient excitations in the sample across the Fermi energy which equilibrate to a Fermi-Dirac distribution on a timescale of about 100 fs[26,27]. The material dependent characteristics of the electron cascade is the main cause of radiation damage[28,29]. The transiently stored energy in the electrons is transferred to the lattice on a longer timescale of about 1 ps owing to the large difference in specific heat of the two systems.

While the interaction of high intensity X-ray pulses with matter is a relatively new area of study[1,2,12,15,30–32], it appears that after the electron cascade trickles down to energies of a few eV, X-ray-excited samples reach states that are similar to those reached through optical excitation, and can be expected to exhibit some of the same dynamics. The fluence-dependent XAS changes seen in Fig. 3 are strongly reminiscent of XAS changes seen in a previous study following optical excitation of nickel[33], as well as changes in extreme ultraviolet emission of extreme ultraviolet excited aluminum[2]. Those changes have been described well as redistribution of valence electrons from below to above the Fermi level[2,34,35], although a potential change in electron localization has also been discussed[33]. The valence electron redistribution results in an increased XAS signal below $E_{core \rightarrow Fermi}$ due to the creation of holes below the Fermi level. Similarly, the decrease in XAS signal above $E_{core \rightarrow Fermi}$ results from an increase in electron population above the Fermi level that reduces the excitation probability.

The interpretation of the fluence-dependent XAS changes due to a valence electron redistribution simply follows from a

description of the 3d density of states in an independent electron model[36,37] and application of Fermi's golden rule for the dependence of transition probabilities on the final state occupation. Within this model, the location of $E_{\text{core}\rightarrow\text{Fermi}}$ in Figs. 2 and 3 corresponds to the zero crossing of the fluence-dependent changes in XAS. The XAS changes in Fig. 3 are approximately equal below and above $E_{\text{core}\rightarrow\text{Fermi}}$ indicating that the total number of 3d holes in Co is conserved during the X-ray pulse. This shows that there is no significant charge transfer between Co and Pd. The XAS changes are well represented by an electronic temperature change within a rigid band model. The shown fit is for an electronic temperature change from 300 K to 3300 K with a Lorentzian broadening of 430 meV FWHM to account for the Co $L_3$ lifetime and a Gaussian broadening of 260 meV FWHM to account for experimental resolution[38]. That the XAS changes are well represented by this electronic temperature change indicates that there are no significant changes of the electronic density of states within the X-ray pulse duration. It is remarkable that it is possible to model the transient electron redistribution averaged over our pulse length of 39 fs by a temperature dependent Fermi-Dirac distribution which is expected to be applicable only at longer timescales of about 100 fs[26,27] corresponding to the establishment of an equilibrium.

As the electronic excitations created by X-ray absorption equilibrate they also impact the spin-dependent structure of the material. This happens in different ways at different stages of equilibration. The high energy electrons (>30 eV), which are created following X-ray absorption and subsequent inelastic electron scattering, do not scatter in a spin-dependent manner. In other words, high energy, spin-polarized electrons produce an equal number of lower energy secondary electrons per primary electron[39]. These electrons, however, may contribute to spin transport in the material, and thus, an apparent demagnetization of the Co component of the sample[40]. In particular, due to their greater number, more majority than minority electrons are excited in Co through inelastic scattering of high energy electrons, and these electrons may then travel to the Pd regions of the sample that our X-ray measurements are not sensitive to. Once electrons reach lower energies of several eV or less, they are in similar states as can be reached with direct optical excitation and will contribute to demagnetization in the same ways as for optically-induced demagnetization. The mechanisms driving ultrafast optically induced demagnetization[41] are a matter of ongoing debate[40,42–44]. Spin flip as well as spin transport processes have both been observed to be significant with their relative importance depending on the investigated sample and its geometry[43,45]. Recent experimental studies have hinted at the importance of magnons[46–48] and understanding the first 30 fs after electronic excitation[43,49,50]. In addition, previous XMCD measurements have revealed different responses of spin and orbital moments on femtosecond timescales[51]. In our experiment, however, we are not sensitive to different dynamics of spin and orbital moments as we measure XMCD changes only at the $L_3$ resonance.

There are several effects which have been observed in experiment or commonly proposed to occur with ultrafast demagnetization and which may impact the spin-dependent unoccupied density of states, and thus our nonlinear XMCD results. First, there may be changes in the spin polarization at particular energies due to spin flip transitions or spin transport at those energies[52]. This would cause changes in the spin polarization at energies where there are significant amounts of both unoccupied and occupied states (within 1 eV of $E_{\text{core}\rightarrow\text{Fermi}}$ in our case). Second, there may be spin-dependent changes in the occupations of states near the Fermi level due to a redistribution of spin-dependent carriers[46,48,50]. This would also cause changes in spin

polarization at energies where there are excited carriers. Redistribution of spin-dependent carriers, however, would not change the total spin polarization. Third, there may be a change in exchange splitting[26,49]. A change of the exchange splitting could change the spin-dependent density of states in a complex manner, although the unoccupied states may not be strongly affected[46]. A change in exchange splitting would also not, by itself, change the total spin polarization. Fourth, the inhomogeneity of the sample magnetization may increase through magnon generation[46,48]. This quenches long range magnetic order, while short range magnetic order and exchange splitting remain. For measurements such as ours that average over a macroscopic sample area, the result is that one measures the same spin averaged density of states while the measured energy dependence of the majority and minority states become mixed, an effect that has been called band mirroring[48]. Band mirroring has also been seen in temperature-dependent measurements of magnetic structure[53,54].

The change of XMCD shown in Fig. 3b is consistent with a uniform reduction in the XMCD within the error of the measurement. Of the above listed effects, only band mirroring due to magnon generation and spin flip or spin transport at specific energies contribute to a net reduction of the XMCD. Spin flip or spin transport at specific energies, however, does not, by itself, change the XMCD at photon energies where there are not a significant amount of both occupied and unoccupied states to excite into. In contrast, the XMCD changes at photon energies well above $E_{\text{core}\rightarrow\text{Fermi}}$, even where the XAS is negligibly changed. This shows that band mirroring is the dominant of these contributions to the overall reduction in XMCD. We note, however, that this does not necessarily mean this is the first step in the demagnetization. Redistribution of spin-dependent carriers and changes of exchange splitting may additionally change the shape of the XMCD reduction, particularly near $E_{\text{core}\rightarrow\text{Fermi}}$. While the fit to a uniform reduction in XMCD is not as good near $E_{\text{core}\rightarrow\text{Fermi}}$, the deviation was not strong enough for us to definitively conclude that additional effects are important in its description. A change in exchange splitting, in particular, would be difficult to detect as this may not impact the spin-dependent unoccupied states greatly[46], and reported changes in exchange splitting have typically been only a few hundred meV or less[26,49], while our ability to resolve changes in the spin-dependent unoccupied states is limited to 430 meV by the Co $2p_{3/2}$ core hole lifetime[55].

The dominance of band mirroring in the changes of the spin-dependent density of states during demagnetization is consistent with optically induced demagnetization of Co on an insulating substrate probed with extreme ultraviolet magneto-optical measurements of Co on an insulating substrate[46,47], and time- and spin-resolved photoemission measurements of optically-induced demagnetization of Co on Cu[48]. [48] additionally observed a deviation from a purely band mirroring model near the Fermi level that they attributed to a redistribution of spin-polarized carriers. In contrast,[50] observed that, upon optical excitation, the spin polarization near the Fermi level of Fe on W decreases within 60 fs while the band-mirroring effect has a longer timescale of about 450 fs. These differences could be due to the different samples used in the experiments (Co/Pd multilayers in our study, Co on an insulating substrate in[46,47], Co on Cu in ref. [48], but Fe on W in ref. [50]).

Figure 4 shows a quantification of the X-ray absorption changes with respect to pulse averaged absorbed X-ray energy density. The vertical error bars of Fig. 4 are standard errors of the displayed quantities. The horizontal error bars correspond to the estimated 20% uncertainty in the calibration of the absolute X-ray fluence (the relative uncertainty for different X-ray fluence points

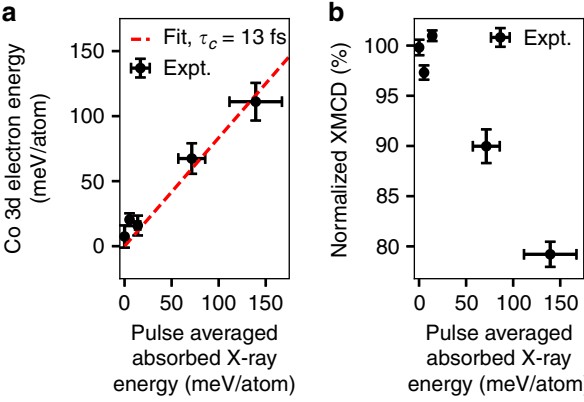

**Fig. 4** Quantification of X-ray absorption changes. The quantities derived from the X-ray absorption changes are plotted as a function of the pulse averaged energy absorbed in the Co/Pd multilayer (one half of the total absorbed energy). Vertical error bars correspond to the standard error of the displayed quantities while horizontal error bars correspond to the estimated 20% uncertainty in the absolute fluence calibration. **a** Co 3d electron energy averaged over the X-ray pulse as calculated from the fluence-dependent XAS changes. The linear fit to the data (dashed red line) has a slope of 0.83 indicating that 83% of the pulse averaged absorbed X-ray energy is stored by 3d states within 2 eV of the Fermi level (averaged over the pulse duration). Comparing this value to a simple model for the transfer of absorbed X-ray energy to electrons within 2 eV of the Fermi level, we estimate the duration of this cascade process to be $\tau_c = 13$ fs. **b** Normalized XMCD strength averaged over its FWHM extent in photon energy. Source data provided as a Source Data file.

is much less). The data extends up to a pulse averaged absorbed X-ray energy of 140 meV/atom.

Figure 4a shows an estimation of the valence excitation strength as a function of the pulse averaged deposited X-ray energy using the XAS spectra and the XAS sum rule. This rule states that $N_h = k[L_3 + L_2]$, where $N_h$ is the number of Co 3d holes per atom, $L_3$ and $L_2$ are the integrals over the resonant components of the Co $L_3$ and $L_2$ edges, and $k$ is a proportionality constant[9]. As we measured XAS changes only at the Co $L_3$ resonance, we approximated this sum rule as $N_h = mL_3$, where $m$ is another proportionality constant. To calculate $m$, we took $N_h$ to be 2.49 and subtracted a non-resonant background from the low fluence X-ray absorption spectra recorded at a synchrotron light source, as in ref. [56]. The change in number of Co 3d holes per unit energy is approximately given by an energy dependent version of this sum rule, $\Delta N_h(E) = m\Delta L_3(E)$, where $E$ is the particular energy of interest, and $\Delta L_3(E)$ is the change in XAS at that energy. The change in Co 3d electron energy per atom, $\Delta V$, is then given by multiplying by the distance below $E_{\text{core}\rightarrow\text{Fermi}}$ and integrating over energy:

$$\Delta V = -m \int (E - E_{\text{core}\rightarrow\text{Fermi}})\Delta L_3(E)dE. \qquad (1)$$

Performing this integration from 2 eV below to 2 eV above $E_{\text{core}\rightarrow\text{Fermi}}$, we obtain the results shown in Fig. 4a. The data is well represented by a linear fit with the spectrally calculated valence electron energy being 83% of the total deposited X-ray energy. This represents the degree to which the deposited X-ray energy has been transferred to electrons within 2 eV of the Fermi level within the X-ray pulse duration.

We further used the spectrally detected valence electron energy as a function of pulse averaged absorbed X-ray energy to estimate the duration of the cascade from initial high energy X-ray excitation to valence excitations within 2 eV of the Fermi level. To do

so, we calculated, within a simple model and for the parameters of this experiment, the cascade duration, $\tau_c$, which gives a spectrally detected electron energy within 2 eV of the Fermi level which is 83% of the pulse averaged absorbed X-ray energy, as for the fit to the data of Fig. 4a (see methods). The result is a cascade duration of $\tau_c = 13$ fs. This estimated X-ray-induced electron cascade duration is in reasonable agreement with the predicted duration of transfer of electronic energy from initial several hundred eV excitations to excitations with energies of 10 eV or less following soft X-ray excitation of condensed matter[22,57].

Figure 4b shows the XMCD as a function of the absorbed X-ray energy per atom. The strength of the XMCD decreases with increasing absorbed X-ray energy. For the highest absorbed energy density case of 140 meV per atom pulse averaged absorbed X-ray energy, the XMCD has decreased in magnitude by 21%. In comparison, using the microscopic three temperature model[42], we estimate that the measured degree of optically induced demagnetization in a similar experiment but with optical light would be 23% for Co/Pt magnetic multilayers and 8% for Co (see methods, Supplementary Table 1 and Supplementary Fig. 2). Thus, the 21% X-ray-induced demagnetization that we observe here is consistent with that which would be expected for optical demagnetization of similar materials with otherwise the same experimental conditions.

In summary, we have presented X-ray fluence-dependent changes of X-ray absorption at the Co $L_3$ edge of Co/Pd magnetic multilayers. The spectral changes reflect a transfer of deposited X-ray energy to valence electrons within 2 eV of the Fermi level in <20 fs and demagnetization of >20%. Our study shows that X-ray-induced dynamics can be used to gain new insights into ultrafast processes, such as ultrafast demagnetization. In other cases, it is desirable to use high intensity X-ray pulses to probe sample characteristics without modification of the sample properties through X-ray excitation[58,59]. Our study sets limits on when this is possible.

## Methods

**Fabrication and magnetization of Co/Pd samples**. The Co/Pd samples were sputter deposited onto 100 nm thick $Si_3N_4$ membranes. They had a metal layer sequence of Ta(1.5)Pd(3)[Co(0.6)Pd(0.6)]x38Pd(2), where the thicknesses in parentheses are in nm. During measurement, the samples were magnetized out-of-plane with an applied magnetic field of 350 mT. This field was sufficient to saturate the magnetization out-of-plane, as verified through hysteresis loop measurements (see Supplementary Fig. 3).

**X-ray magnetic circular dichroism measurements**. We performed this experiment at the SXR hutch[60] of the LCLS X-ray free electron laser[58,61]. The Delta undulator was operated in the diverted beam scheme, providing circularly polarized X-ray pulses with 200 μJ pulse energy and 25 fs FWHM pulse duration[62]. A grating monochromator then filtered these X-rays to a bandwidth of 260 meV and broadened the pulses by 34 fs FWHM[38] (P. Heimann, personal communication) resulting in a pulse length of 39 fs FWHM (as estimated numerically). After monochromatization, the gas monitor detector measured the absolute X-ray pulse energy[63]. A fluorescence-based X-ray detector then measured the shot-to-shot incoming pulse energy[17], before the X-rays traversed a variable attenuation solid attenuator system, which enabled adjustment of the X-ray fluence at the sample. Next, a pair of Kirkpatrick–Baez mirrors focused the X-rays onto the sample at normal incidence with a spot size of 11 by 21 μm FWHM, as measured by pinhole scans. The X-rays transmitted through the sample were attenuated before being detected with a CCD (Andor DO940P-BN-T2). The CCD was operated in spectroscopy mode, where the signal is integrated over each pixel column of the CCD before read out. In this mode, the CCD could be read out at the full 120 Hz repetition rate of LCLS. When data was recorded with fluences near and above the threshold for permanent sample damage, samples were replaced at a rate of 2 Hz. In calculating spectra, we only used data recorded on samples which had not been exposed to fluences above a threshold well below that required for permanent X-ray absorption changes (see Supplementary Fig. 4 for plots of permanent X-ray absorption changes induced by exposure to high X-ray fluences).

**Calculation of X-Ray absorption spectra**. Synchrotron light source spectra were collected over a 100 eV range in electron yield mode. The photon energy was calibrated so that the peak of the Co $L_3$ resonance occurred at 778 eV, while that of

the Co $L_2$ resonance occurred at 793.2 eV. A linear background was then fit to the pre-edge of the recorded spectra and subtracted in order to get the calibrated spectra. The intensity of the synchrotron light source spectra were scaled so that they aligned with that recorded at the XFEL with 0.01 mJ/cm$^2$ incident fluence.

The XFEL spectra were recorded in transmission and had a linear background subtracted. All XFEL spectra used the same photon energy calibration determined by aligning the lowest fluence XFEL spectrum (0.01 mJ/cm$^2$) with that recorded at the synchrotron light source. For a set of spectra recorded with a specific incident fluence, the $\sigma_-$ and $\sigma_+$ spectra were recorded simultaneously, and thus had the same linear background subtracted. The background was determined such that the spectra align with the low fluence limit for photon energies well away from the absorption resonance (773.5 eV to 774.5 eV and 781 eV to 782 eV). From previous studies, it has been seen that any spectral changes occurring so far away from the resonance are much less than that occurring within a few eV of the resonance[33,51]. As a confirmation of the validity of this procedure, the 19 mJ/cm$^2$ and 43 mJ/cm$^2$ spectra of Fig. 2, which were recorded under the same conditions except with different incident X-ray fluences, have the same linear background subtracted and do not exhibit spectral changes well away from the Co $L_3$ absorption resonance. Before calculation of spectra, the response of the incident X-ray detector was calibrated with data recorded without a sample to correct for a small saturation at high X-ray pulse energies. See Supplementary Fig. 1 for further details.

**Determination of absolute pulse energy.** The absolute X-ray pulse energy at the sample was determined using the gas monitor detector upstream of the sample and the CCD downstream of the sample. Despite the fundamental physical differences in the operation of these detectors, they gave values for the absolute pulse energy at the sample which were within 20 percent of each other. In calculating the pulse energy at the sample, the efficiency or transmission of components between the pulse energy detector and the sample was taken into account. The transmission of attenuators inserted before or after the sample was measured. The efficiency of each of the soft X-ray mirrors was estimated to be 0.73, and the transmission of the SiN membrane used in the $I_0$ detector was calculated to be 0.59 for the measured photon energies[19].

**Melting Threshold of Co/Pd.** For solids, permanent sample damage is usually observed for samples exposed to single-shot X-ray doses larger than the melting threshold[64,65]. The threshold dose for melting of the Co/Pd multilayer samples, $D_{melt}$, was approximated as

$$D_{melt} = f_{Co}\left[H_{f,Co} + H_{Co}(T_{melt}) - H_{Co}(T_0)\right] + f_{Pd}[H_{Pd}(T_{melt}) - H_{Pd}(T_0)], \quad (2)$$

where $f_x$ is the fraction of atomic species x in the Co/Pd multilayers, $H_{f,Co}$ is the enthalpy of fusion of Co, $H_x(T)$ is the enthalpy of x at temperature $T$, and $T_{melt}$ is the melting temperature of Co, which has a lower melting temperature than Pd[20]. Using the melting temperatures and atomic densities given in[20], as well as the enthalpies given in[66,67], we obtain $D_{melt} = 0.615$ eV/atom. The enthalpies tabulated in[67] have been interpolated to $T_{melt}$ where necessary.

**Model of X-ray-induced electron cascade.** We used a simple model to estimate the duration of the X-ray-induced electron cascade based on the estimated Co 3d electron energy per atom shown in Fig. 4a. The electronic energy within 2 eV of the Fermi level was given by $\nu(t) = (I * c)(t)$, where $*$ denotes convolution, $c(t)$ gives the fraction of energy transferred to within 2 eV of the Fermi level following instantaneous X-ray excitation at $t = 0$, and $I(t)$ is the intensity of the X-ray pulse at the sample. $c(t)$ was modeled as a linear increase from zero at $t < 0$ to one at $t > \tau_c$, with $\tau_c$ being the duration of the electron cascade. $I(t)$ was modeled as the convolution of a flattop function with 25 fs FWHM (representing the pulse produced by LCLS) and a Gaussian with 34 fs FWHM (representing the pulse broadening due to the monochromator). We note that this neglects the spiky temporal structure of the XFEL pulses which were produced through self-amplified spontaneous emission[68]. The spectrally detected valence electron energy, s, was then given by averaging the valence electron energy over the X-ray pulse in time, $s = (\int dt I(t)\nu(t))/(\int dt I(t))$. Using this model, we estimate an X-ray-induced electron cascade duration of 13 fs for evolving from initial absorption of an X-ray photon to excitations within 2 eV of the Fermi level.

**Microscopic three temperature model simulations.** To compare the pulse-averaged X-ray-induced demagnetization and electronic temperature to that which would be expected for similar experiments with optical or infrared light, we simulated these quantities for such experiments. We performed these simulations within the Microscopic Three Temperature Model (M3TM)[42]. We note that the limits of validity of M3TM are a matter of current investigation[50,69] particularly for timescales as short as we examine here, and thus, the demagnetization magnitudes we extract here should only be seen as a rough approximation of the true demagnetization that would occur. The equations governing the evolution of a

sample within M3TM and neglecting spatial dependence for simplicity are[42,70]

$$C_e \frac{dT_e}{dt} = g_{ep}(T_p - T_e) + P_{pump} \quad (3)$$

$$C_p \frac{dT_p}{dt} = g_{ep}(T_e - T_p) \quad (4)$$

$$\frac{dm}{dt} = Rm\frac{T_p}{T_C}\left(1 - m\coth\left(\frac{mT_C}{T_e}\right)\right). \quad (5)$$

Here, $T_e$ is the electronic temperature, $T_p$ is the lattice temperature, and $P_{pump}$ is the energy density deposited per unit time in the sample. $C_e$ is the temperature-dependent electronic heat capacity given by $C_e = \gamma T_e$, where $\gamma$ is a materials-dependent parameter. $C_p$ is the lattice heat capacity, $g_{ep}$ is the electron–phonon coupling, $R$ is a materials-specific scaling factor for the demagnetization rate, $m = M/M_s$ is the magnetization relative to its value at zero temperature, and $T_C$ is the Curie temperature.

We chose to simulate the optical pump-induced dynamics for Co/Pt multilayers and Co as these samples are similar to the Co/Pd multilayers we studied in the X-ray experiment, and the M3TM parameters for Co and Co/Pt are available in the literature. These parameters are shown in Supplementary Table 1. Supplementary Fig. 2 shows the results of simulating sample evolution within M3TM for Co and Co/Pt samples excited with optical pulses. We used the same FWHM (39 fs) and total deposited energy per atom (280 meV/atom), as the X-ray pulses used in Fig. 3. The simulated pulse-averaged demagnetization for Co was 8%, while that for Co/Pt was 23%. The 21% X-ray-induced pulse-averaged demagnetization we measured for Co/Pd falls between these values. The simulated pulse-averaged electronic temperature was 2340 K, while that for Co/Pt was 2090 K. These values are somewhat less than the 3300 K electronic temperature obtained by fitting the XAS of Fig. 3b.

## Data availability

Raw data were generated at the LCLS large-scale facility. Derived data supporting the findings of this study are available from the corresponding authors upon request. The source data underlying Figs. 2–4 and Supplementary Figs. 1–4 are provided as a Source Data file.

## Code availability

Code used to carry out the analyses in this manuscript is available on GitHub (github.com/dhigley6/nonlinear-xmcd).

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

## Acknowledgements

We acknowledge M. Beye, C. D. Pemmaraju, D. A. Reis, and P. Heimann for valuable discussions. We thank J. Aldrich for technical assistance. Use of the Linac Coherent Light Source, SLAC National Accelerator Laboratory, is supported by the U.S. Department of Energy, Office of Science, Office of Basic Energy Sciences under Contract No. DE-AC02-76SF00515. The Advanced Light Source is supported by the Director, Office of Science, Office of Basic Energy Sciences, of the U.S. Department of Energy under Contract No. DE-AC02-05CH11231. L.L.G. would like to thank the VolkswagenStiftung for financial support through the Peter-Paul-Ewald Fellowship.

## Author contributions

D.J.H., A.H.R., Z.C., L.L.G., A.A.L., T.L., P.S., T.C., G.L.D., A.M., E.Y., J.S., H.A.D., W.F.S., and J.S. performed experiments. O.H. and Z.C. prepared samples. D.J.H. analyzed experimental data and performed calculations. D.J.H., A.H.R., and J.S. wrote the manuscript with input from all authors.

## Competing interests

The authors declare no competing interests.
