## [Peer review file · Nature Communications]

Reviewers' Comments:

Reviewer #1:

Remarks to the Author:

In the manuscript "Femtosecond X-Ray Induced Changes of the Electronic and Magnetic Response of Solids From Electron Redistribution", Higley et al. investigated and reported the X-ray absorption spectroscopy (XAS) and X-ray circular dichroism (XMCD) measurement of Co/Pd multilayer with a range of X-ray light fluence. From the results, the authors obtained several very interesting observations: 1) The high-energy excited electrons cascaded down to 2eV around the Fermi energy in a surprisingly short timescale (<40 fs at the beginning of the manuscript and $\tau_c \sim 12$ fs at the end); 2) The XMCD response of the sample depends on the X-ray fluence, which can be explained as the X-ray-induced demagnetization, similar to the optically induced demagnetization. Ultrafast laser-induced demagnetization has been a topic of interest for many years. So far most of the results were obtained either with near-IR light excitation or with terahertz light excitation, while the effect of the X-ray light on material magnetization is a mostly unexplored region. The results presented in this work will shed some light on this new direction. Meanwhile, the fluence-dependent results in this work are new, representing a nonlinear response of the magnetic materials to strong X-ray pulse, and X-ray-induced magnetic dynamics. So, I think this work is very interesting, and will be an important contribution to the community.

Meanwhile, I feel especially glad to see that the authors carefully investigated and found that the transient electron redistribution of photoexcited electrons can already be fit by a temperature dependent Fermi-Dirac distribution within 40fs. In my previous studies, my colleagues and I have got a similar conclusion in our previous experiments and was very surprised that it is much shorter than the 100 fs timescale from previous works. I am very glad to see that this can be confirmed independently using a different method.

Even though I give the above appraisal to this work, I still have a few questions and comments to this work. I will recommend the manuscript to be published in Nature Communications, only after the authors can address them.

1) From the beginning of the manuscript (page 4), the authors started to compare the deposited X-ray energy density with the melting dose. I don't quite understand the intention behind this comparison, because, for both the nonlinear optical spectroscopy or the ultrafast demagnetization, the melting dose of 615 meV per atom is a very large value. In most of the experiments, the deposited energy is a way below this limit. For example, in Ref. [47], Tengdin et al. estimated that the energy required for melting the magnetic order is only 105 meV per unit cell. So, the authors should explain in more details about their intention of this comparison.

2) Related to my comment 1), at the end of the manuscript, the authors concluded that the highest fluence corresponds to a 130 meV per atom absorbed X-ray energy, which leads to $\sim 22\%$ of the demagnetization amplitude. It is great to have this number, but how does it compare to the previous demagnetization experiments excited by NIR pulses? Is there any difference? Without such discussion, I just found this value reported here lacks its significance.

3) The authors believe that the XMCD change can be explained by the band-mirroring model of Ref. [46]. I, however, think this argument lacks sufficient support from the experimental data. The authors made this conclusion based on two observations: a) The uniform reduction in XMCD response as a function of fluence, and b) the "possible" deviation of the model to the experimental data around the Fermi energy, which is shown in Fig. 3b. First of all, I failed to see a convincing difference between the experiment and fit in Fig. 3b. The authors should put the error bars on the experimental data in order to compare with the fit result. Meanwhile, the authors used "possibly" a couple of times in their description, so I believe they are also not entirely sure about the validity of this evidence. Regarding the evidence a), its validity depends on the energy resolution and the ability to detect the details of the band structure in the XAS and XMCD experiments, for which the

authors should provide more details in the paper.

4) Overall, the authors have described interesting electronic and magnetic responses after the magnetic sample was excited by strong X-ray pulses. However, the manuscript has been organized in a way that these two parts of the results were intertwined in the text, but lack connection in logic. From the experimental results and models, can the authors provide a clearer physical picture of the electronic and magnetic dynamics after X-ray-pulse excitation?

Reviewer #2:

Remarks to the Author:

The paper presents X-ray Absorption Spectroscopy (XAS) experimental results obtained with Free Electron Laser pulses with increasing X-ray fluence and discuss the origin of the modifications observed with respect to "usual" synchrotron radiation sources. A model is proposed which well reproduces XAS and X-ray magnetic circular dichroism (XMCD) measured at the Co L3 absorption edge.

The results are important for a wide scientific community, they are well presented and discussed. The authors should consider the following comments before publication.

1) The discussion of the results is strongly dependent on the relative intensity of the XAS spectra measured with synchrotron and FEL radiation. Even if the complex procedure needed to extract the XAS spectra from FEL measurements is described in the text as well as the alignment and normalization procedures, the raw data presentation, at least in supplementary information is important for the understanding of the process needed to obtain the usual I/I₀ signal.

2) In the discussion of the observed fluence dependent XAS changes the authors introduce the location of the Fermi level in XAS spectra, which is placed at the zero crossing of the measured curves. If I can understand the interest of using this approach to discuss "the absorption changes dominated by valence electronic and magnetic changes of the sample itself", the text oversimplify the problem, there is no Fermi level to be found in XAS spectra and the term should not be used in this way for XAS. The model based on modified density of states can be used to explain the experimental results but using the correct terms to describe electronic excitations from the core level to empty valence states.

3) Interestingly, XMCD signal changes only at photon energies higher than the zero crossing point, but this is not discussed in the manuscript.

Reviewer #3:

Remarks to the Author:

The authors investigated the effects of femtosecond X-Ray pulses on the XAS and XMCD of Co L3 edge of Co/Pd multilayer metal films with a metal layer sequence of

Ta(1.5)Pd(3)[Co(0.6)Pd(0.6)]x38Pd(2), where the thicknesses are in parentheses are in nm. They concluded that the sample demagnetizes by more than twenty percent, subjected to the femtosecond X-Ray pulses. The energy-dependence of demagnetization reflects a mixing of the density of states for majority and minority electrons. This experiment and the discussions can be referred by relative investigations. The present manuscript can be accepted to publish.

However, if the following suggestions can be adopted by the authors, the revision will be better than present manuscript.

1. Add a monolayer Co film sample, and compare the results with the present sample.
2. Present the magnetic loops of the samples before and after subjected to femtosecond X-Ray pulses.

We have detailed the reviewer feedback along with our point-by-point responses below. In addition, we streamlined the analysis procedure to produce the results of this manuscript so that all the calibration procedures are done automatically. This simplified analysis produced the same results within the experimental error. We uploaded the analysis code and the data necessary to reproduce the results described in this manuscript to Github (<https://github.com/dhigley6/nonlinear-xmcd>).

Reviewer #1:

In the manuscript “Femtosecond X-Ray Induced Changes of the Electronic and Magnetic Response of Solids From Electron Redistribution”, Higley et al. investigated and reported the X-ray absorption spectroscopy (XAS) and X-ray circular dichroism (XMCD) measurement of Co/Pd multilayer with a range of X-ray light fluence. From the results, the authors obtained several very interesting observations: 1) The high-energy excited electrons cascaded down to 2eV around the Fermi energy in a surprisingly short timescale (<40 fs at the beginning of the manuscript and $\tau_c \sim 12$ fs at the end); 2) The XMCD response of the sample depends on the X-ray fluence, which can be explained as the X-ray-induced demagnetization, similar to the optically induced demagnetization. Ultrafast laser-induced demagnetization has been a topic of interest for many years. So far most of the results were obtained either with near-IR light excitation or with terahertz light excitation, while the effect of the X-ray light on material magnetization is a mostly unexplored region. The results presented in this work will shed some light on this new direction. Meanwhile, the fluence-dependent results in this work are new, representing a nonlinear response of the magnetic materials to strong X-ray pulse, and X-ray-induced magnetic dynamics. So, I think this work is very interesting, and will be an important contribution to the community.

Meanwhile, I feel especially glad to see that the authors carefully investigated and found that the transient electron redistribution of photoexcited electrons can already be fit by a temperature dependent Fermi-Dirac distribution within 40fs. In my previous studies, my colleagues and I have got a similar conclusion in our previous experiments and was very surprised that it is much shorter than the 100 fs timescale from previous works. I am very glad to see that this can be confirmed independently using a different method.

Even though I give the above appraisal to this work, I still have a few questions and comments to this work. I will recommend the manuscript to be published in Nature Communications, only after the authors can address them.

1) From the beginning of the manuscript (page 4), the authors started to compare the deposited X-ray energy density with the melting dose. I don't quite understand the intention behind this comparison, because, for both the nonlinear optical spectroscopy or the ultrafast demagnetization, the melting dose of 615 meV per atom is a very large value. In most of the experiments, the deposited energy is a way below this limit. For example, in Ref. [47], Tengdin et al. estimated that the energy required for melting the magnetic order is only 105 meV per unit cell. So, the authors should explain in more details about their intention of this comparison.

The intention in making this comparison is a practical one. While the sample melting dose sets one limit on the maximum permissible X-ray fluence in most experiments with XFELs on solids, our results show that the maximum permissible fluence can be lower due to a nonlinear sample response.

In order to make the significance of this comparison clear, we have edited the text of the manuscript to have a single reference to the sample melting threshold at the end of the results section. That text now reads

The results of Fig. 3 show that for experiments that use femtosecond X-ray pulses to measure the linear, un-damaged response of solids to X-ray pulses the maximum permissible X-ray fluence can be remarkably low. The resonant X-ray absorption of Co/Pd changes by more than ten percent with a total deposited X-ray energy density of 280 mV/atom. Thus, even for experiments which seek to measure linear resonant X-ray absorption with an accuracy of only ten percent, the deposited X-ray energy density should be much less than 280 meV/atom. This dose is much lower than the estimated sample melting dose of 615 meV/atom (see methods), which sets another limit on the maximum permissible X-ray fluence in cases where samples are not refreshed between measurements.

2) Related to my comment 1), at the end of the manuscript, the authors concluded that the highest fluence corresponds to a 130 meV per atom absorbed X-ray energy, which leads to ~22% of the demagnetization amplitude. It is great to have this number, but how does it compare to the previous demagnetization experiments excited by NIR pulses? Is there any difference? Without such discussion, I just found this value reported here lacks its significance.

We have added a comparison to the degree of demagnetization expected for similar experiments with similar materials, but with NIR light rather than X-ray pulses. In the main text, we have added the following sentences

In comparison, using the microscopic three temperature model[42], we estimate that the measured degree of optically-induced demagnetization in a similar experiment but with optical light would be 23 percent for Co/Pt magnetic multilayers and 8 percent for Co (see supplementary material). Thus, the 21 percent X-ray-induced demagnetization that we observe here is consistent with that which would be expected for optical demagnetization of similar materials with otherwise the same experimental conditions.

In addition, we have added Supplementary Note 1 and Supplementary Figure 4 that describe the calculation of these values.

3) The authors believe that the XMCD change can be explained by the band-mirroring model of Ref. [46]. I, however, think this argument lacks sufficient support from the experimental data. The authors made this conclusion based on two observations: a) The uniform reduction in XMCD response as a function of fluence, and b) the “possible” deviation of the model to the experimental data around the Fermi energy, which is shown in Fig. 3b. First of all, I failed to see a convincing difference between the experiment and fit in Fig. 3b. The authors should put the error bars on the experimental data in order to compare with the fit result. Meanwhile, the authors used “possibly” a couple of times in their description, so I believe they are also not entirely sure about the validity of this evidence. Regarding the evidence a), its validity depends

on the energy resolution and the ability to detect the details of the band structure in the XAS and XMCD experiments, for which the authors should provide more details in the paper.

We have added error bars to the data of Fig. 3b to compare with the fit result. While the fit is not as good near the Fermi level as elsewhere, the differences near the Fermi level are not large enough to definitively conclude that there has been a change in the XMCD lineshape. We have changed the language used in the manuscript in correspondence with this. We also changed the discussion of the potential contributions to the XMCD to reflect this and added a more detailed accounting of the contributions and potential contributions to the XMCD change. The changes are copied below

There are several effects which have been observed in experiment or commonly proposed to occur with ultrafast demagnetization and which may impact the spin-dependent unoccupied density of states, and thus our nonlinear XMCD results. First, there may be changes in the spin polarization at particular energies due to spin flip transitions or spin transport at those energies [52]. This would cause changes in the spin polarization at energies where there are significant amounts of both unoccupied and occupied states (within 1 eV of $E_{\text{core} \rightarrow \text{Fermi}}$ in our case). Second, there may be spin-dependent changes in the occupations of states near the Fermi level due to a redistribution of spin-dependent carriers [46, 48, 50]. This would also cause changes in spin polarization at energies where there are excited carriers. Redistribution of spin-dependent carriers, however, would not change the total spin polarization. Third, there may be a change in exchange splitting [26, 49]. A change of the exchange splitting could change the spin-dependent density of states in a complex manner, although the unoccupied states may not be strongly affected [46]. A change in exchange splitting would also not, by itself, change the total spin polarization. Fourth, the inhomogeneity of the sample magnetization may increase through magnon generation [46, 48]. This quenches long range magnetic order, while short range magnetic order and exchange splitting remain. For measurements such as ours that average over a macroscopic sample area, the result is that one measures the same spin averaged density of states while the measured energy dependence of the majority and minority states become mixed, an effect that has been called band mirroring [48]. Band mirroring has also been seen in temperature-dependent measurements of magnetic structure [53, 54].

The change of XMCD shown in Fig. 3B is consistent with a uniform reduction in the XMCD within the error of the measurement. Of the above listed effects, only band mirroring due to magnon generation and spin flip or spin transport at specific energies contribute to a uniform reduction of the XMCD. Spin flip or spin transport at specific energies, however, does not, by itself, change the XMCD at photon energies where there are not a significant amount of both occupied and unoccupied states to excite into. In contrast, the XMCD changes at photon energies well above $E_{\text{core} \rightarrow \text{Fermi}}$, even where the XAS is negligibly changed. This shows that band mirroring is the dominant of these contributions to the overall reduction in XMCD. We note, however, that this does not necessarily mean this is the first step in the demagnetization. Redistribution of spin-dependent carriers and changes of exchange splitting may additionally change the shape of the XMCD reduction, particularly near $E_{\text{core} \rightarrow \text{Fermi}}$. While the fit to a uniform reduction in XMCD is not as good near $E_{\text{core} \rightarrow \text{Fermi}}$, the deviation was not strong enough for us to definitively conclude that additional effects

are important in its description. A change in exchange splitting, in particular, would be difficult to detect as this may not impact the spin-dependent unoccupied states greatly [46], and reported changes in exchange splitting have typically been only a few hundred meV or less [26, 49], while our ability to resolve changes in the spin-dependent unoccupied states is limited to 430 meV by the Co $2p_{3/2}$ core hole lifetime [55].

The dominance of band mirroring in the changes of the spin-dependent density of states during demagnetization is consistent with extreme ultraviolet magneto-optical measurements of Co on an insulating substrate [46, 47], and time- and spin-resolved photoemission measurements of Co on Cu [48]. [48] additionally observed a deviation from a purely band mirroring model near the Fermi level that they attributed to a redistribution of spin-polarized carriers. In contrast, [50] observed that, upon optical excitation, the spin polarization near the Fermi level of Fe on W decreases within 60 fs while the band-mirroring effect has a longer timescale of about 450 fs. These differences could be due to the different samples used in the experiments (Co/Pd multilayers in our study, Co on an insulating substrate in [46, 47], Co on Cu in [48], but Fe on W in [50]).

4) Overall, the authors have described interesting electronic and magnetic responses after the magnetic sample was excited by strong X-ray pulses. However, the manuscript has been organized in a way that these two parts of the results were intertwined in the text, but lack connection in logic. From the experimental results and models, can the authors provide a clearer physical picture of the electronic and magnetic dynamics after X-ray-pulse excitation?

The electronic and magnetic dynamics are co-occurring and closely connected. The ultrafast demagnetization of the sample is a result of the equilibration of the high energy electronic excitations created through X-ray absorption. To emphasize this link, and form a clearer picture of the electronic and magnetic dynamics we have changed the beginning of the discussion section which describes the magnetic dynamics to

As the electronic excitations created by X-ray absorption equilibrate they also impact the spin-dependent structure of the material. This happens in different ways at different stages of equilibration. The high energy electrons (> 30 eV), which are created following X-ray absorption and subsequent inelastic electron scattering, do not scatter in a spin-dependent manner. In other words, high energy, spin-polarized electrons produce an equal number of lower energy secondary electrons per primary electron [39]. These electrons, however, may contribute to spin transport in the material, and thus, an apparent demagnetization of the Co component of the sample [40]. In particular, due to their greater number, more majority than minority electrons are excited in Co through inelastic scattering of high energy electrons, and these electrons may then travel to the Pd regions of the sample that our X-ray measurements are not sensitive to. Once electrons reach lower energies of several eV or less, they are in similar states as can be reached with direct optical excitation and will contribute to demagnetization in the same ways as for optically-induced demagnetization. The mechanisms driving ultrafast optically-induced demagnetization [41] are a matter of ongoing debate [40, 42–44]. Spin flip as well as spin transport processes have both been observed to be significant with their relative importance depending on the

investigated sample and its geometry [43, 45]. Recent experimental works have hinted at the importance of magnons [46–48] and understanding the first 30 fs after electronic excitation [43, 49, 50]. In addition, previous XMCD measurements have revealed different responses of spin and orbital moments on femtosecond timescales [51]. In our experiment, however, we are not sensitive to different dynamics of spin and orbital moments as we measure XMCD changes only at the L_3 resonance.

Reviewer #2:

The paper presents X-ray Absorption Spectroscopy (XAS) experimental results obtained with Free Electron Laser pulses with increasing X-ray fluence and discuss the origin of the modifications observed with respect to “usual” synchrotron radiation sources. A model is proposed which well reproduces XAS and X-ray magnetic circular dichroism (XMCD) measured at the Co L_3 absorption edge.

The results are important for a wide scientific community, they are well presented and discussed. The authors should consider the following comments before publication.

1) The discussion of the results is strongly dependent on the relative intensity of the XAS spectra measured with synchrotron and FEL radiation. Even if the complex procedure needed to extract the XAS spectra from FEL measurements is described in the text as well as the alignment and normalization procedures, the raw data presentation, at least in supplementary information is important for the understanding of the process needed to obtain the usual I/I₀ signal.

We have added Supplementary Figure 2 to illustrate the analysis procedure.

2) In the discussion of the observed fluence dependent XAS changes the authors introduce the location of the Fermi level in XAS spectra, which is placed at the zero crossing of the measured curves. If I can understand the interest of using this approach to discuss “the absorption changes dominated by valence electronic and magnetic changes of the sample itself”, the text oversimplify the problem, there is no Fermi level to be found in XAS spectra and the term should not be used in this way for XAS. The model based on modified density of states can be used to explain the experimental results but using the correct terms to describe electronic excitations from the core level to empty valence states.

We have changed the terminology we use to be more clear. We have added the following sentences to reflect this

The correct photon energy for exciting Co $2p_{3/2}$ core electrons to into unoccupied states at the Fermi level is estimated as the zero crossing of the change in XAS, between where the change in XAS is positive below the absorption resonance, and negative at the peak of the absorption resonance. The reasoning for this is described in the discussion section below. We refer to this position as $E_{\text{core} \rightarrow \text{Fermi}}$, and its position is indicated in Fig. 2 with dashed vertical lines at 777.5 eV. The incident X-ray photon energy is shown on the bottom x-axes

of Fig. 2, while the photon energy above $E_{\text{core} \rightarrow \text{Fermi}}$ is shown on the top x-axes. Above $E_{\text{core} \rightarrow \text{Fermi}}$, the incident X-rays have sufficient photon energy to excite Co $2p_{3/2}$ core electrons into unoccupied valence states.

We also refer to the zero crossing of the measured XAS curves as $E_{\text{core} \rightarrow \text{Fermi}}$ where it occurs elsewhere in the text.

3) Interestingly, XMCD signal changes only at photon energies higher than the zero crossing point, but this is not discussed in the manuscript.

This point is closely related to point 3 of reviewer #1. A lack of an XMCD change below the zero crossing point would be an indication of a change of the lineshape of the XMCD. While the fit to the XMCD change shown in Fig. 3b (representing a uniform reduction of the XMCD) is below the measured XMCD changes below the zero crossing point, we concluded that this deviation was not far enough from what would be expected given the statistical error to definitively conclude that the XMCD changed less below the zero crossing point.

Reviewer #3:

The authors investigated the effects of femtosecond X-Ray pulses on the XAS and XMCD of Co L3 edge of Co/Pd multilayer metal films with a metal layer sequence of Ta(1.5)Pd(3)[Co(0.6)Pd(0.6)]x38Pd(2), where the thicknesses are in parentheses are in nm. They concluded that the sample demagnetizes by more than twenty percent, subjected to the femtosecond X-Ray pulses. The energy-dependence of demagnetization reflects a mixing of the density of states for majority and minority electrons. This experiment and the discussions can be referred by relative investigations. The present manuscript can be accepted to publish.

However, if the following suggestions can be adopted by the authors, the revision will be better than present manuscript.

1. Add a monolayer Co film sample, and compare the results with the present sample.

We thank the reviewer for this interesting suggestion for potential future experiments. In our study, our sample is of a sufficient thickness that the soft X-ray-excited electrons largely do not escape the sample. In contrast, for a Co monolayer, a significant fraction of excited electrons may leave the sample. The spectroscopic changes we observe may therefore be significantly reduced in the case of a monolayer sample. Further, the electronic and magnetic characteristics of monolayer Co are markedly different from that of bulk Co, and this would likely also impact the X-ray-induced dynamics. Unfortunately, the required sensitivity for such an experiment is much greater than the one we describe in our manuscript, and is likely not accessible with our present instrumentation. This is because the X-ray absorption intensity is proportional to the thickness of the sample, and thus, the X-ray absorption intensity for a monolayer would be ~100 times less than that of the Co/Pd samples we used. Further, conducting such an experiment requires a

successful application for beamtime at a highly competitive X-ray free electron laser. We therefore relegate such investigations to future reports, but thank the reviewer for the suggestion.

2. Present the magnetic loops of the samples before and after subjected to femtosecond X-Ray pulses.

We recorded a magnetic hysteresis loop of one of the samples prior to exposure to femtosecond X-ray pulses, and this is now included as Supplementary Figure 1. We did not record the same kind of hysteresis loop on a sample after exposure to femtosecond X-ray pulses. We did, however, record XMCD spectra after exposing the samples to femtosecond X-ray pulses of varying fluences. We have included a figure of this data as Supplementary Figure 3.

Reviewers' Comments:

Reviewer #1:

Remarks to the Author:

The authors have made extensive edits in the revised manuscript. These edits have addressed most of my comments in my review report. I don't have further questions or concerns to this manuscript and would like to recommend this paper publishing in Nature Communications.

Reviewer #2:

Remarks to the Author:

The author's answers to the referee comments are convincing. The clear description added in the supplementary information well explains the calibration procedure used. The manuscript presentation of the absorption process avoids now misunderstandings and presents the experimental results in an objective way. The results and the physics are easier to read and to understand. It will be an important reference for the magnetization dynamics community. I recommend the manuscript for publication.

Reviewer #3:

Remarks to the Author:

The authors have responded the comments of reviewers. The present manuscript can be accepted to publish.

We have detailed the reviewer feedback below. The reviewers did not request any additional changes to our manuscript. We thank the reviewers for their positive appraisals and constructive criticism of our manuscript.

Reviewer #1:

The authors have made extensive edits in the revised manuscript. These edits have addressed most of my comments in my review report. I don't have further questions or concerns to this manuscript and would like to recommend this paper publishing in Nature Communications.

Reviewer #2:

The author's answers to the referee comments are convincing. The clear description added in the supplementary information well explains the calibration procedure used. The manuscript presentation of the absorption process avoids now misunderstandings and presents the experimental results in an objective way. The results and the physics are easier to read and to understand. It will be an important reference for the magnetization dynamics community. I recommend the manuscript for publication.

Reviewer #3:

The authors have responded the comments of reviewers. The present manuscript can be accepted to publish.